# dFCExpert: Learning Dynamic Functional Connectivity Patterns with Modularity and State Experts

## Abstract

Modeling brain dynamic functional connectivity (dFC) patterns from functional Magnetic Resonance Imaging (fMRI) data is of paramount importance in neuroscience and medicine. Recently, many graph neural networks (GNN) models in conjunction with transformers or recurrent neural networks (RNNs) have been proposed and shown great potential for modeling dFC patterns in terms of pattern recognition and prediction performance. Although fruitful, several issues still hinder further performance improvement of these methods, such as neglecting the intrinsic brain modularity mechanism, and the interpretable state information of dFC patterns. To tackle these limitations, we propose dFCExpert to learn effective representations of dFC patterns in fMRI data with modularity experts and state experts. Particularly, using the GNN and mixture of experts (MoE), the modularity experts characterize the brain modularity organization in the graph learning process by optimizing multiple experts, with each expert capturing brain nodes with similar functions (in the same neurocognitive module); and the state experts aggregate temporal dFC features into a set of distinctive connectivity states by a soft prototype clustering methods, where the states can support different brain activities or are affected differently by brain disorders, thus revealing insights for interpretability. Experiments on two large-scale fMRI datasets demonstrate the superiority of our method over known alternatives, and the learned dFC representations show improved explainability and hold promise to improve clinical diagnosis.

## 1 Introduction

The human brain is a dynamic network system that generates complex spatiotemporal dynamics of brain activity. Analyzing such dynamics holds promising to provide insights into the brain's functional organization and its relationship with human cognition (Greicius, 2008; Wang et al., 2007), behaviors (Smith et al., 2015), and brain disorders (Greicius, 2008; Wang et al., 2007). Among many in-vivo brain imaging techniques, functional Magnetic Resonance Imaging (fMRI) is a powerful means that models the spatiotemporal patterns of brain activity by measuring fluctuations in blood-oxygen level-dependent (BOLD) signals (Matthews & Jezzard, 2004). Since brain activity exhibits strong spatial correlations, the BOLD signals generally are summarized into a collection of pre-defined brain regions (ROIs), and the pairwise correlations between those ROIs are referred as functional connectivity (FC), which has emerged as a key tool for understanding the brain function.

Based on the FC, the brain can be modeled as a functional network using graph theory approaches, with a set of ROIs as the network nodes and the connectivity strengths of FC as edges. Thus, due to this graph-structured nature of the brain, many recent methods have employed graph neural networks (GNNs) to learn intricate brain network representations, and then explore tasks such as decoding human traits or diagnosing diseases (Li et al., 2021; Kawahara et al., 2017; Kan et al., 2022). Generally, these models have two distinct lines: *static FC* and *dynamic FCs methods*. The *static-FC methods* measure the FC between nodes based on the entire fMRI scan, i.e., the full time series (Ktena et al., 2017; 2018; Kim & Ye, 2020). However, these studies fail to take advantage of the dynamic properties of the FCs (fluctuate over time), which are considered essential for capturing the brain's evolving states. Differently, the *dynamic-FCs methods* split the whole fMRI time series into temporal segments and quantify the FC based on each segment so that time-varying FC

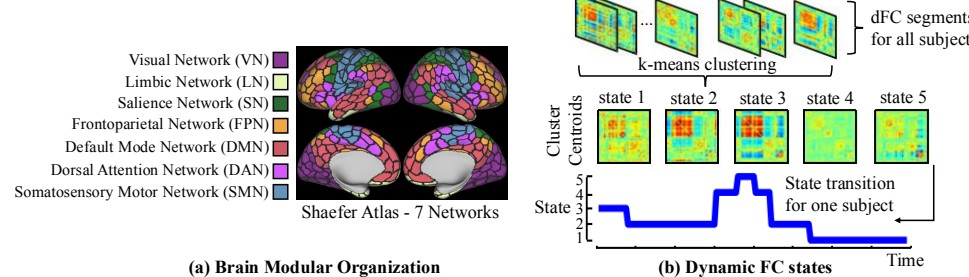

Figure 1: (a) The Shaefer atlas (Schaefer et al., 2018) with 7 functional modules. Each black line marked region indicates a node and the same color regions indicate one functional module. (b) dFC states are a small number of patterns of dFC measures, summarizing how the brain FCs evolving.

measures can be derived. From this, a universal pipeline for the dynamic-FCs methods is to extract brain network representations using GNNs for each FC segment, and then use RNNs or transformers for temporal dynamics learning (Kim et al., 2021; Wang et al., 2023; Jing et al., 2023; Lin et al., 2023). Although effective, these methods still exhibit several issues, primarily due to the unique characteristics of brain functional networks and dFC measures.

The first issue is that most GNN methods for brain functional network analysis neglect the inherent brain modularity mechanism, resulting in suboptimal brain graph representation learning. Theoretical and empirical studies have revealed that the human brain works as a modular system with specialized cognitive/topological modules, each consisting of brain nodes (ROIs) tightly connected and together responsible for certain functions (Sporns & Betzel, 2016; Bertolero et al., 2015), as shown in Figure 1(a). However, known GNN-based methods process all brain nodes similarly (share the same aggregation mechanism), regardless of the differences in the node features, which hinder them from capturing modularity specific features. The second issue is that known studies also ignore the interpretable dFC state patterns. The dFC measures can capture different dynamic states of the brain by clustering dFC measures of temporal segments of fMRI scans. Therefore, the dFC states can compactly describe and summarize how the brain FCs vary temporally (Figure 1(b)), and provide interpretable information. Studies of brain disorders have shown that disease-specific alterations only happen on some of the dynamic states (Damaraju et al., 2014), indicating that capturing dFC states can potentially improve the detection of functional brain changes caused by brain disorders.

To tackle these issues, we propose a novel GNN-based dFC learning framework called dFCExpert, aiming to enhancing the representation learning of dynamic functional connectome of fMRI data. Concretely, the framework consists of modularity experts and state experts, specifically designed following the brain modularity characteristic and the dFC state patterns. The modularity experts are the conjunction of GNN and Mixture of Experts (MoE) for learning brain graph features of each FC segment, which characterize the brain modularity organization in the graph learning process by optimizing multiple aggregation experts at each graph layer, with each expert focusing on specific subsets of nodes with akin behaviors and interactions. On top of the learned graph representations from modularity experts, the state experts aggregate the temporal features of dFCs into a smaller set of distinctive states based on a soft prototype clustering method, where each state has similar FC patterns and reflects different activities of the dFCs related to human behaviors or brain diseases. Finally, the clustering-derived state features are used for task prediction. With the modularity and state experts, our dFCExpert explores two novel strategies for learning-based fMRI analysis (i.e., the brain modularity and dynamic state mechanisms in neuroscience), which presents a first-of-its-kind method for learning informative and explainable features for dFC patterns from fMRI data.

We evaluate the performance of dFCExpert on two large fMRI datasets: the Human Connectome Project (HCP) (Van Essen et al., 2013) and the Adolescent Brain Cognitive Development (ABCD) (Casey et al., 2018). Across the sex classification and cognitive intelligence prediction tasks, dFC-Expert significantly outperforms known methods and achieves state-of-the-art performance. We comprehensively analyze individual components of dFCExpert and conduct ablation studies to substantiate our modeling choices. Visualization results show that our modularity experts can specialize in particular brain modules and the state experts present interpretable evidence for prediction tasks.

We also demonstrate that the brain modules and states identified by our method are biologically meaningful, providing a bridge to connect deep learning based functional network findings with conventional functional network analytic methods.

## 2 RELATED WORKS

**Dynamic Brain Connectome Learning**     To capture the time-varying FC patterns of the brain, one very beginning method is based on the spatio-temporal graph convolutional network (ST-GCN) (Gadgil et al., 2020), which incorporates spatio-temporal convolution to model the non-stationary nature of functional connectivity. Then, many methods follow a GNN-RNN pipeline to learn graph-level features for each FC segment with the GNNs and then capture temporal FC patterns with recurrent neural networks (RNNs) or transformers. Particularly, STAGIN (Kim et al., 2021) utilizes GNNs with spatial attention and transformers with temporal attention to model the dynamics of brain networks. DynDepNet (Campbell et al., 2022) proposes to learn the time-varying structures of fMRI data through a dynamic graph structure learning method. NeuroGraph (Said et al., 2023) presents a collection of graph-based neuroimaging datasets and has systematically studied the effectiveness of various GNN designs on dynamic brain graph data. Although these methods have demonstrated the potential to capture the dynamics of brain connectivity, they ignore the inherent brain modular structures and are not equipped to capture dynamic state patterns of dFCs, which impedes the provision of insights into the functional organization and connectivity states of the brain.

**Brain Modularity Organization**     The human brain operates as a modular system with many cognitive/topological modules, and brain regions (ROIs) in the same modules are often densely connected and tend to have similar functions. For example, salience network (SN) and default mode network (DMN) are two crucial neurocognitive modules in the brain, where SN mainly detects internal or external stimuli and coordinates the brain's response to those stimuli, and DMN is responsible for self-related cognitive functions (Sporns & Betzel, 2016; Bertolero et al., 2015). A few existing methods have incorporated the brain modular organization in the brain graph representation learning process. BRAINNETTF (Kan et al., 2022) designs a novel readout function to take advantage of the modular-level similarities between ROIs, where the graph-level embeddings are pooled from clusters of functionally similar nodes. MSGNN (Wang et al., 2023) proposes a modularity-constrained GNN to learn graph features, and it constrains the node embeddings into 3 neurocognitive modules (i.e., central executive network, SN, DMN) by encouraging similarity between node-level embeddings in the same module. However, Since these methods only consider the brain modularity at a late stage of their graph models, all brain network nodes are still similarly processed in each graph layer (i.e., all nodes share the same aggregation mechanism). Such a strategy might be suboptimal for effectively learning representations of nodes belonging to distinct brain modules. Thus, we propose modularity experts, the conjunction of GNN and MoE, to imitate the brain's modular organization, which comprise multiple aggregation experts at each graph layer and brain nodes with similar functions are guided towards the same experts during training.

**Dynamic FC States**     The dynamic brain states are a small number of unique patterns of dFC measures. To extract interpretable information from the dFC patterns, neuroscience researches often obtain dynamic FC states by summarizing the dFC patterns into a smaller set of connectivity states, which can represent patterns of connectivity that repetitively occur during the fMRI acquisition (Preti et al., 2017). Existing studies typically cluster the dFC measures into different states using clustering methods (e.g., k-means clustering) and then carry out statistic analyses to explore statistical relation between the states and biological or behavior measures of interest (Damaraju et al., 2014). However, such a two-step strategy is not able to effectively identify informative dFC states since features of dFC graphs used for the identification of dFC states and the statistical analyses are unnecessarily the same. In contrast, our state experts are learning-based, and learn distinctive dFC states and informative representations of dFC graphs in an end-to-end fashion.

**Expert Models**     The concept of Mixture of Experts (MoE) (Jacobs et al., 1991) has a long history as a machine learning technique that employs multiple expert layers, each specializing in solving a specific subtask or learning a specific sub-structure. Recently, the remarkable success of MoEs has facilitated its application in various domains, such as multi-modal learning (Akbari et al., 2024; Mustafa et al., 2022), vision (Jain et al., 2024; Riquelme et al., 2021), and machine translation (Shen et al., 2019). When it comes to graph classification, TopExpert (Kim et al., 2023) has adopted

Figure 2: dFCExpert consist of modularity experts and state experts. The modularity experts have $L$ layers of MoE-GIN, which route node features $h_i$ to a specific GIN expert with a gating score $G(h_i)$, then obtaining graph-level feature $\mathbf{h}_{modularity} = (\mathbf{h}_{G_1}, \ldots, \mathbf{h}_{G_T})$. The state experts aggregate the learned graph features into different dynamic state patterns by prototype gating and embedding projection. Finally, the state features $\mathbf{h}_{state}$ with an MLP layer are used for task prediction.

the MoEs on top of the extracted graph features, and leveraged topology-specific prediction experts for molecule property prediction. Instead of applying MoEs after a GNN extractor, (Wang et al., 2024) proposes a graph MoE (GMoE) model by adopting multiple experts in each graph layer to scale GNN model's capacity. Our dFCExpert is distinguished from the above approaches in that we aim to learn brain modularity and dFC state representations from dynamic graphs.

## 3 METHOD

### 3.1 PRELIMINARIES

**Graph Neural Networks** To encode useful information from graph-structured data, GNNs iteratively compute node features by aggregating information of neighbor nodes and updating the node features with non-linear functions in a layer-wise manner. The propagation mechanism of node $i$ at $(l)$-th GNN layer can be formulated as:

$$h_i^{(l)} = F_u^{(l)}\left(h_i^{(l-1)}, F_a^{(l)}\left(\{(h_i^{(l-1)}, h_j^{(l-1)}, e_{ij}) : j \in N_i\}\right)\right), \tag{1}$$

where $h_i^{(l-1)}$ denotes the features for node $i$ at layer $(l-1)$, $N_i$ is the set of neighbors of node $i$, $e_{ij}$ is the edge between node $i$ and node $j$, and $F_a$ and $F_u$ are differentiable functions for aggregating information and updating representations, respectively. Generally, different choices of these functions can yield distinctive variants of the GNN. For example, Graph Isomorphism Network (GIN) (Xu et al., 2018a), a variant of GNN, uses sum as $F_a$ and a multi-layer perceptron (MLP) as $F_u$:

$$h_i^{(l)} = M^{(l)}(h_i^{(l-1)}, e_{ij}, W^{(l)}) = \text{MLP}^{(l)}\left((1 + \epsilon^{(l)}) \cdot h_i^{(l-1)} + \sum_{j \in N_i} h_j^{(l-1)}\right). \tag{2}$$

We simplify the above two functions as $M^{(l)}$, where $W^{(l)}$ is the trainable weight in MLP and $\epsilon$ is a learnable parameter initialized with zero. Due to its powerful ability for graph representation learning, we use GIN as our brain graph feature extractor as in (Kim et al., 2021).

**Dynamic Graph Definition** To construct dynamic FC graphs from fMRI scans, we first use a brain atlas to convert the 4D fMRI data into a time-course matrix $\mathbf{P} \in \mathbb{R}^{N \times T_{\max}}$, which is extracted by taking the mean values of fMRI signals within each region of the brain atlas, containing $N$ brain network nodes (ROIs) in each timepoint. After that, following the sliding-window approach, the time-course matrix is divided into $T = \lfloor T_{\max} - \Gamma/S \rfloor$ temporal segments with a window length of $\Gamma$ and a stride size of $S$. We compute FC matrix by calculating Pearson correlation coefficients between time series of pairwise ROI for each of T segments, resulting in $X_t \in \mathbb{R}^{N \times N}(t = 1, \ldots, T)$. As shown in (Said et al., 2023), the correlation matrices can be informative node features, where the features for $i$-th node are elements of the $i$-th row in $X_t$ for segment $t$. A binary adjacent

matrix $A_t \in \{0,1\}^{N \times N}$ for segment $t$ is generated from the FC matrix $X_t$, by retaining the top 30-percentile values of the correlation as connected, and others as unconnected following (Kim & Ye, 2020). Thus, the input dynamic FC graphs for each subject is $\mathbf{G}_t = \{A_t, X_t\}(t = 1, \ldots, T)$.

## 3.2 DFCEXPERT

### 3.2.1 OVERVIEW

As illustrated in Figure 2, dFCExpert consist of modularity experts and state experts. Taking the dynamic FC graphs as input, the modularity experts learn brain graph features for each FC segment by combing GNN and MoE, with the goal of characterizing the brain modularity mechanism. On top of the modularity experts, the state experts adaptively group the temporal graph features into distinctive states based on a soft prototype clustering method, where expressive state features can be learned by assigning soft clusters to the temporal graph features. Finally, the learned state features with an MLP layer are used to predict a specific task, such as predicting sex in a classification setting or intelligence measure in a regression setting. Formally, the goal of our dFCExpert is to train a neural network $f : (\mathbf{G}_1, \ldots, \mathbf{G}_T) \rightarrow \mathbf{h}_{state}$, where $\mathbf{G}_t = \{A_t, X_t\}(t = 1, \ldots, T)$ is the sequence of our constructed FC graphs with $T$ segments and $\mathbf{h}_{state} \in \mathbb{R}^{K \times D}$ is the output feature of the state experts. Going one step further, we formulate $f = m \circ s$ as a composition of modularity experts $m$ and state experts $s$, where $m$ learns the brain graph representations $\mathbf{h}_{modularity} = (\mathbf{h}_{G_1}, \ldots, \mathbf{h}_{G_T})$ for each of $T$ segments and $s$ aggregates these temporal graph features into the state feature $\mathbf{h}_{state}$:

$$m : (\mathbf{G}_1, \ldots, \mathbf{G}_T) \rightarrow (\mathbf{h}_{G_1}, \ldots, \mathbf{h}_{G_T}), \qquad s : (\mathbf{h}_{G_1}, \ldots, \mathbf{h}_{G_T}) \rightarrow \mathbf{h}_{state}. \qquad (3)$$

Following, we will elaborate the proposed modularity experts and state experts.

### 3.2.2 MODULARITY EXPERTS

The modularity experts imitate the human brain's modular organization to learn effective brain graph representations. We implement our modularity experts using GIN in conjunction with MoE, denoted as MoE-GIN, and we have $L$ layers of MoE-GIN. Specifically, each MoE-GIN layer comprises a gating function and multiple GIN experts, as shown in Figure 2. The gating function determines which experts to use for a given node, and the multiple experts are independent GIN node propagation functions, each having its own trainable parameters. This design of MoE-GIN allows tightly connected or behaviorally similar nodes to be assigned to the same experts, thereby enabling each GIN expert to specialize in a particular brain module and capture specific brain activities. In this way, the modularity experts work just like a brain, where densely connected nodes interact together to perform a specific function or activity. Formally, a MoE-GIN layer is described as:

$$h_i^{(l)} = \sum_{c=1}^{C} G_c^{(l)}(h_i^{(l-1)}) M_c^{(l)}(h_i^{(l-1)}, e_{ij}, W^{(l)}) \qquad (4)$$

where $C$ is the number of experts, and $M$ denotes the node propagation function of GIN as in Eq (2). $G$ is the gating function that produces the assignment scores with the input of node feature $h_i^{(l-1)}$, and $G_c(h_i^{(l-1)})$ indicates the $c$-th score. We implement a simple gating function by multiplying the input with a trainable weight matrix $W_g$ and then applying the $\mathrm{softmax}$ function (Jordan & Jacobs, 1994): $G(h_i) = \mathrm{softmax}(h_i W_g)$. Considering that each node generally belongs to one single brain module, we employ a simplified strategy for $G$ with the top-1 gating scheme following (Fedus et al., 2022), where each node only routes to a single expert. This simplification can reduce the routing computation without compromising model quality. We compute the graph-level representation $h_{G_t}^{(l)}$ for segment $t$ at layer $l$ by averaging the updated node features $\{h_i^{(l)}, i \in N\}$, and the final graph representation $\mathbf{h}_{G_t}$ is a concatenation of graph representations of all $L$ layers (Xu et al., 2018b) followed by an MLP layer for dimension reduction: $\mathbf{h}_{G_t} = \mathrm{MLP}(\mathrm{concatenate}(h_{G_t}^{(l)}|l \in \{1, \cdots, L\})) \in \mathbb{R}^D$. Finally, we obtain $\mathbf{h}_{modularity} = (\mathbf{h}_{G_1}, \ldots, \mathbf{h}_{G_T}) \in \mathbb{R}^{T \times D}$.

**Auxiliary Loss Functions for Training MoE-GIN** However, if the model is trained solely using the task-specific prediction loss, the gating network tends to converge to a trivial solution where only a few experts are consistently selected (Shazeer et al., 2017). This issue of imbalanced selection is self-reinforcing, since the favored experts can proliferate more rapidly than others, resulting

in their increased frequency of selection. Another trivial situation is that the gating network also tends to generate similar expert assignment scores for a given node, resulting in no specialization in experts. In our scenario, we expect the gating network to be capable of distinguishing nodes of different brain functional modules and enable sparse gating, where the chosen expert could have a high score than others. To help the training of MoE-GIN, we present two auxiliary loss functions to encourage good loading balance and sparse gating, respectively. Given a batch $B$, each with $T$ graphs, the loading balance loss is computed as:

$$L_{balance} = \frac{1}{BTC} \sum^B \sum^T \sum_{c=1}^C p_c \log(p_c), \qquad p_c = \frac{1}{N} \sum_{i=1}^N G_c(h_i),$$ (5)

where $N$ is the number of nodes and $p_c$ is the fraction of these nodes dispatched to expert $c$. By minimizing the negative entropy of the $p_c$ distribution, the $L_{balance}$ can allow nodes uniformly distributed on experts. To facilitate the sparse gating of $G(h_i)$, we design a sparse loss:

$$L_{sparse} = \frac{1}{BTN} \sum^B \sum^T \sum^N \frac{1}{C} \sum_{c=1, g \in G(h_i)}^C -g_c log(g_c).$$ (6)

Minimizing the sum of gating scores' entropy over all experts encourages experts to have sparse scores for each node. We implement these two losses similarly as in (Huang et al., 2020), since they are actually deep clustering-based losses and are against each other during the training process. For each MoE-GIN layer, these two auxiliary losses are added to the total model loss during training.

### 3.2.3 STATE EXPERTS

The state experts follow the typical dFC analysis in neuroscience to adaptively group the temporal graph features $\mathbf{h}_{modularity}$ into a small number of states, each reflecting different dynamic states of the dFCs related to human behaviors or brain diseases. From another respective, it is also much easier to capture the FC dynamics from a smaller set of FC states, and it can simplify the model and reduce the complexity of the data. Specifically, we implement the state experts based on a soft prototype clustering method, where distinctive dFC states are characterized by a prototype gating and embedding projection as illustrated in Figure 2. The prototype gating introduces $K$ trainable prototype centroids $U = [u_1, \ldots, u_K]$, each having $D$ dimensions ($U \in \mathbb{R}^{K \times D}$), to adaptively learn feature centers of each state. Given the learned graph features $\mathbf{h}_{modularity} = (\mathbf{h}_{G_1}, \ldots, \mathbf{h}_{G_T})$, the prototype gating first calculates the score $P_{tk}$ for assigning segment $t$ to state $k$ by a Softmax projection:

$$P_{tk} = \frac{e^{\langle \mathbf{h}_{G_t}, u_k \rangle}}{\sum_{k'}^K e^{\langle \mathbf{h}_{G_t}, u_{k'} \rangle}},$$ (7)

where $\langle \cdot, \cdot \rangle$ indicates the inner product. Then, the embedding projection aggregates the temporal graph features $\mathbf{h}_{modularity} \in \mathbb{R}^{T \times D}$ into state features $\mathbf{h}_{state}$ under the guidance of the obtained soft assignment score $P \in \mathbb{R}^{T \times K}$. Particularly, to compute features for one state (e.g., State 1 as in Figure 2), the embedding projection performs $P_{:1}^\top \mathbf{h}_{modularity}$ to integrate features of the temporal segments assigned to State 1 with the soft gating. Thus, the state features can be obtained by aggregating the temporal FC features with high probability assigned to a specific state and sharing similar dFC patterns. To achieve better gating results, the initialization of $K$ prototype centroids is important, and good performance can be achieved by encouraging the orthogonality of the centers (Kan et al., 2022). We use orthonormal initialization and more details are in Appendix Section A.

## 4 EXPERIMENTS

### 4.1 EXPERIMENTAL SETTINGS

**Datasets** We conduct experiments on two real-world fMRI datasets. (a) *Human Connectome Project (HCP)*: We use the dataset from HCP S1200 release (Van Essen et al., 2013), which consists of pre-processed and ICA denoised (Glasser et al., 2013) resting-state fMRI scans of 1071 individuals with 578 females and 493 males. Following (Kim et al., 2021), we use data from the first run and exclude data with short acquisition time (fewer than 1200 time series), and all the used resting-state

Table 1: Performance comparison with alternatives and baselines. ("M"= Modularity, "S" = State)

| Method | HCP | | | | ABCD | | | | FC Type |
| | Sex | | Intelligence | | Sex | | Cognition | | |
| | ACC(%) | AUC(%) | MSE(↓) | CORR(↑) | ACC(%) | AUC(%) | MSE(↓) | CORR(↑) | |
|---|---|---|---|---|---|---|---|---|---|
| BrainNetCNN (Kawahara et al., 2017) | 84.56 | 91.97 | 1.130 | 0.200 | 85.38 | 92.67 | 0.474 | 0.442 | Static |
| BRAINNETTF (Kan et al., 2022) | 85.42 | 92.51 | 1.015 | 0.205 | 84.55 | 91.38 | 0.462 | 0.465 | Static |
| NeuroGraph (Said et al., 2023) | 84.38 | 92.03 | 1.321 | 0.198 | 85.71 | 93.75 | 0.512 | 0.423 | Static |
| STAGIN (Kim et al., 2021) | 87.74 | 92.09 | 1.002 | 0.215 | 86.93 | 91.57 | 0.452 | 0.477 | Dynamic |
| MSGNN (Wang et al., 2023) | 86.25 | 92.37 | 1.073 | 0.206 | 87.13 | 92.67 | 0.453 | 0.483 | Dynamic |
| NeuroGraph (Said et al., 2023) | 84.66 | 93.14 | 1.017 | 0.207 | 87.37 | 94.64 | 0.441 | 0.482 | Dynamic |
| Baseline | 88.11 | 96.68 | 0.978 | 0.238 | 87.25 | 94.86 | 0.442 | 0.484 | Dynamic |
| Baseline + M | 89.61 | 96.91 | 0.946 | 0.272 | 88.09 | 95.16 | 0.425 | 0.499 | Dynamic |
| Baseline + S | 89.98 | 97.23 | 0.967 | 0.248 | 88.56 | 95.81 | 0.419 | 0.496 | Dynamic |
| dFCExpert | **91.03** | **97.40** | **0.932** | **0.296** | **89.28** | **95.99** | **0.412** | **0.513** | Dynamic |

fMRIs have 1200 timepoints. (b) *Adolescent Brain Cognitive Development Study (ABCD)*: It is a multi-site investigation of brain development and related behavioral outcomes in children aged 9-10 years old (Casey et al., 2018). In our study, we used preprocessed baseline resting-state fMRI data from the ABCD BIDS Community Collection (ABCC) (Feczko et al., 2021). As in (Keller et al., 2023), we exclude participants with incomplete data, excessive head motion, or fewer than 600 remaining timepoints after motion censoring. Finally, after quality control, we use the resting state fMRI scans of 6,165 children, with varied numbers of time points ranging from 626 to 3516.

Both the datasets are publicly available with identification information anonymized. We construct 5 splits (stratified) for both of the HCP and ABCD datasets with a ratio of (train : validation : test) = (7 : 1 : 2), and report the average results across the 5 splits (standard deviation is given in Appendix E).

**Targets and Metrics** We chose classification of sex and prediction of cognitive intelligence ("fluid intelligence" for HCP, and "general cognition" for ABCD) as the evaluation tasks, where the former is a binary classification problem, and the latter is a regression task. These two tasks allow to explore fundamental brain-biology and brain-cognition associations. For the sex classification task, accuracy (ACC) and AUC (Area Under ROC Curve) were used as the evaluation metrics. For the regression task, the regression targets are z-normalized, and Mean Squared Error (MSE) and correlations (CORR) between the measured and predicted values were used to evaluate the model.

**Implementation Details** Our method was implemented using PyTorch (Paszke et al., 2019) and trained on an NVIDIA A100 GPU with 80 GB memory. We set the number of MoE-GIN layers $L = 3$, and embedding dimension $D = 256$. For the dFC graph construction, we used the window length of $\Gamma = 50$ and window stride of $S = 3$, which capture the FC within 36 seconds every 2.16 seconds, following the common protocol of the sliding-window based dFC analyses (Zalesky & Breakspear, 2015; Preti et al., 2017). As in (Kim et al., 2021), we also randomly chose 600 time points for computing the dFC graphs at each training step. This procedure mitigated the computational and memory overhead while augmenting the training data. For testing, we utilized the entire available time-course matrix $\mathbf{P} \in \mathbb{R}^{N \times T_{\max}}$ for dFC graph construction and model evaluation. We train the sex classification and cognitive intelligence regression tasks with cross-entropy and MSE losses, respectively. More implementation details are described in Appendix D.

## 4.2 SEX CLASSIFICATION AND COGNITIVE INTELLIGENCE REGRESSION RESULTS

**Comparison with Known Methods** We compared the performance of dFCExpert with alternative static- and dynamic-FC methods on both sex classification and cognitive intelligence regression tasks. The top two blocks of Table 1 summarize the results of alternative methods. On the sex classification task, dFCExpert outperformed all the existing methods on both HCP and ABCD datasets. On the regression task, our method improved the performance by a large margin, especially on the HCP dataset, with an improvement of nearly 7 points. These quantitative results clearly underscored the necessity and effectiveness of introducing brain modularity and learning state patterns for brain dynamics learning in fMRI data.

**Ablation Studies** We also conducted ablation experiments to evaluate individual components of dFCExpert, namely, the modularity experts and state experts. For fair comparison, the baseline model was a 3-layer GIN with an MLP layer, with the GIN to learn the graph-level features and the MLP to make predictions for each segment $t$ that were averaged to generate the final result. The baseline model was designed for following reasons: 1) Having predictions for each time seg-

Table 2: Performance of the number of experts used in the modularity experts.

| #Experts | HCP | | | | ABCD | | | |
|---|---|---|---|---|---|---|---|---|
| | Sex | | Intelligence | | Sex | | Cognition | |
| | ACC(%) | AUC(%) | MSE($\downarrow$) | CORR($\uparrow$) | ACC(%) | AUC(%) | MSE($\downarrow$) | CORR($\uparrow$) |
| 1 (baseline) | 88.11 | 96.68 | 0.978 | 0.238 | 87.25 | 94.86 | 0.442 | 0.484 |
| 3 | 88.68 | 96.46 | 0.961 | 0.266 | 87.57 | 95.16 | 0.435 | 0.481 |
| 5 | 89.25 | 96.41 | 0.953 | **0.279** | **88.09** | **95.16** | **0.425** | **0.499** |
| 7 | **89.61** | **96.91** | **0.946** | 0.272 | 87.80 | 94.85 | 0.429 | 0.481 |
| 9 | 88.73 | 95.26 | 0.963 | 0.224 | 87.65 | 94.98 | **0.425** | 0.479 |
| 17 | 87.83 | 95.87 | 0.976 | 0.209 | 87.19 | 94.84 | 0.431 | 0.477 |

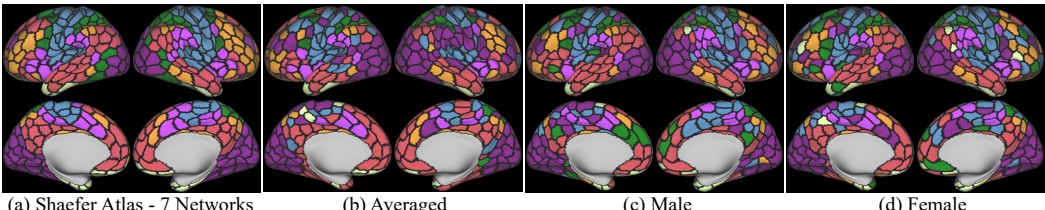

(a) Shaefer Atlas - 7 Networks      (b) Averaged      (c) Male      (d) Female

Figure 3: Visualization of expert assignment results on all subjects (averaged) (b) and two randomly selected subjects (c)(d) on the test set, and comparing with the 7 atlas networks (a). The same color regions of (a) and (b)(c)(d) are the matched results.

ment enables stronger supervision to facilitate improved prediction, compared with the alternatives employing RNNs or transformers on top of the GIN layer (e.g., STAGIN (Kim et al., 2021) and NeuroGraph (Said et al., 2023) in Table 1); 2) Our state experts aggregate the temporal graph features into $K$ state features, and therefore RNNs or transformers are not needed for making the final prediction. Based on the baseline model, we implemented our modularity experts ("Baseline + M") and state experts ("Baseline + S") by replacing the GIN with MoE-GIN and aggregating temporal features before the MLP layer, respectively. The third block of Table 1 summarizes the ablation study results, from which several observations can be drawn: 1) Compared with the baseline model, the modularity experts achieved considerably better performance, especially for the regression task on the HCP dataset, highlighting that characterizing brain modularity was effective for learning informative brain graph features; 2) By grouping the temporal FCs into several dynamic states, better performance was achieved by the state experts on both two tasks and two datasets, highlighting that this new scheme to explore the dynamics of FCs was beneficial; 3) With both the modularity and state experts, our dFCExpert surpassed the baseline model substantially, validating the effectiveness of our method.

### 4.3 MODULARITY EXPERTS ANALYSIS

To demonstrate how the modularity experts affect the performance, we investigated the number of experts used in the MoE-GIN layer. Specifically, we explored the number of experts $C$ within a range of 3, 5, 7, 9, 17. Particularly, $C = 7$ and $C = 17$ are two widely used settings in large-scale functional network studies (Yeo et al., 2011). As summarized in Table 2: 1) On the HCP dataset, the model's performance increased with the increasing of the number of experts from 3 to 7 and then degraded with the number of experts further increased from 7 to 17, suggesting that the expert number should be relatively small, leading to less computation and parameters. 2) On the ABCD dataset, the modularity experts obtained better performance than the baseline setting, and the number of experts had less effect on the classification task while the performance improved a lot on the regression task with the number of experts increasing until $C = 9$. From our results, $C$ can be 5 or 7, which are both reasonable and also consistent with the typical number of functional modules in neuroscience. We finally used $C = 7$ and $C = 5$ for the HCP and ABCD datasets, respectively.

To further evaluate the effectiveness of the modularity experts, we visualized the nodes assignment results by modularity experts with $C = 7$ on the classification task (HCP dataset) and compared it with the Schaefer Atlas (Schaefer et al., 2018) with 7 networks (Figure 3 (a)). That is, we can obtain the 7 expert learned modules, with each containing nodes assigned to specific expert. Since we didn't know how these 7 expert learned modules correspond to the 7 atlas networks, we match

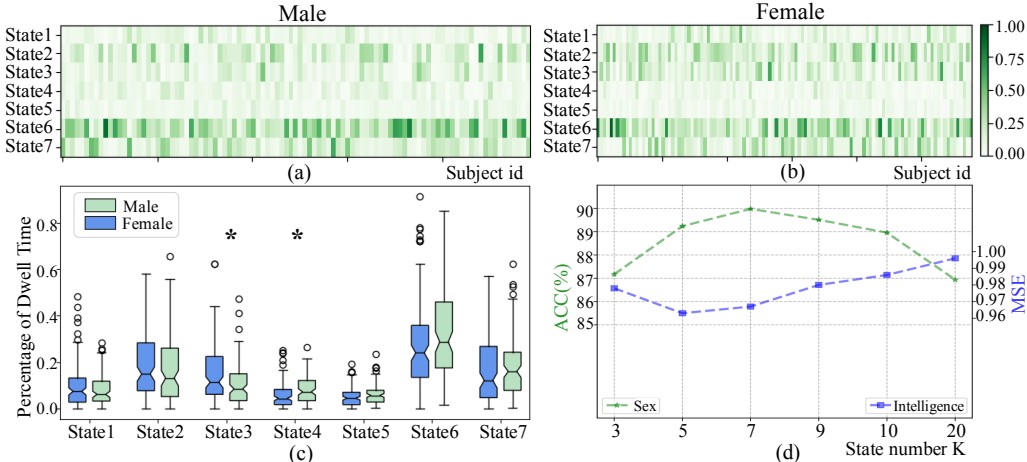

Figure 4: **a)** and **b)** The probabilities of temporal segments of individual males or females assigned to a specific state, where $x$-axis is the individuals in the testing dataset; **c)** Boxplots of a) and b), each box corresponds to one row in a) or b) and ⋆ indicates statistically significant difference; **d)** Performance of different numbers of the dFC states.

them based on the maximum overlapping between their respective nodes, and the matched results were visualized as shown in Figure 3. The visualization clearly demonstrated that the Schaefer atlas networks were highly overlapped with the modules identified by the modularity experts, indicating that the modularity experts could assign tightly connected nodes to the same experts (module) to allow each expert to capture nodes of one brain functional module. Further, since our modularity experts were optimized for specific tasks, they can potentially learn node-expert (or node-network) assignments beyond the prior knowledge of Shaefer Atlas. From Figure 3, we observed individual differences in the resulting learned modules, indicating that the modularity experts had great potential to learn personalized functional networks. Additionally, we also evaluated the effectiveness of the modularity scheme in the static-FC method, analyzed different scaling ratios for the auxiliary losses, and trained the MoE-GIN based on the Schaefer Atlas's network (module) settings. More detailed experimental results can be found in Appendix Section F.

## 4.4 STATE EXPERTS ANALYSIS

To evaluate how the prototype learning for identifying dFC states affects the overall performance, we explored different numbers of states on the HCP dataset and visualized how these states characterized sex differences. Specifically, we investigated the number of states $K$ in a range of 3, 5, 7, 9, 10, 20. Figure 4(d) shows the classification and prediction performance changed with the number of states. Similar to what we observed in the modularity analysis, the performance first improved and then degraded with the increase of the number of states. The best performance was obtained when $K = 7$ on classification task and $K = 5$ on regression task. This was also consistent with the findings in existing dFC studies that the typical number of dynamic states ranges from 5 to 7.

Further, we visualized the state assignment results to evaluate whether the learned states could provide interpretable evidence for distinguishing males from females. Particularly, we first averaged the soft assignment results $P$ across the $T$ dimension so that each resulting value indicated the fraction of the $T$ segments for one subject assigned to specific states (e.g., the $k$-th state). Then, we obtained such values for all subjects in HCP test dataset and visualized them according to males and females separately. The results in Figure 4 (a) and (b) and the boxplots in Figure 4 (c) clearly demonstrated that dFC graphs (temporal segments) of males had higher probabilities to be assigned to States of 4 and 6, while females' dFC graphs were allocated more to States of 2 and 3. Further, the males and females showed significant differences at State 3 and State 4 with p-values less than 0.05 (Wilcoxon rank-sum tests), which validates that the dFC states can show explainable evidence for the sex differences (the males and females have different brain activities and affect different states). Overall, these results clearly demonstrated the effectiveness of grouping dynamic FCs into a smaller set of

dFC states and also illustrated that the state experts can identify distinctive dFC states similar to existing studies, albeit the state experts used a learning strategy different from the conventional dFC studies. More sex difference results are presented in Appendix Section G.

## 5 CONCLUSIONS

We developed a novel method, dFCExpert, for learning effective representations of dFC measures, consisting of the modularity experts and state experts to mirror the brain modularity organization and dFC states that have been extensively investigated in fMRI studies of the functional neuroanatomy, brain development and aging, as well as brain disorders but less exploited in the machine learning community of functional brain networks. As demonstrated by the extensive experimental results on two large scale fMRI datasets, the modularity experts automatically routed network nodes with similar brain activities to the same experts, promoting specialization of each expert for learning effective representations of dFC measures, and the state experts grouped the learned temporal graph features into distinctive states, with each having similar dFC patterns, facilitating effective characterization of the temporal dynamics of the brain functional networks related to different brain states or biological characteristics. These results have demonstrated not only the superior performance of dFCExpert over state-of-the-art alternative methods but also its enhanced interpretability.

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

## A  STATE EXPERTS LEARNING

To help the state experts in terms of clustering, we define a target distribution $Q_{tk}$ for each $\mathbf{h}_{G_t}$ based on the current cluster assignment distribution $P_{tk}$, which therefore enhances the high-confident assignment (Xie et al., 2016). Further, we minimize a KL divergence loss (8) between these two distributions to boost the clustering process. To obtain representative soft assignment $P$, the initialization of $K$ cluster centers is critical, and we use orthonormal initialization as in (Kan et al., 2022).

$$L_{state} = \mathbf{KL}(Q||P) = \frac{1}{T}\sum_{t=1}^{T}\sum_{k=1}^{K} Q_{tk} \log \frac{Q_{tk}}{P_{tk}}, \qquad Q_{tk} = \frac{P_{tk}^2/\sum_t P_{tk}}{\sum_{k'}(P_{tk'}^2/\sum_t P_{tk'})}, \qquad (8)$$

## B  OVERALL LOSS FUNCTION

The final loss employs a task-specific loss ($L_{task}$), two auxiliary losses for MoE-GIN, and one loss for state expert, yielding an overall optimization target ($\alpha$ and $\beta$ are two scaling factors):

$$L_{overall} = L_{task} + \alpha(L_{balance} + L_{sparse}) + \beta L_{state}. \qquad (9)$$

Through experiments of different scaling ratios, we finally set $\alpha = 1$ and $\beta = 10$. Note that the auxiliary losses for modularity experts are the summation of all MoE-GIN layers.

## C  MODEL DETAILS

The modularity Experts are implemented following Neurograph (Said et al., 2023) with a MoE-GIN architecture that incorporates residual connections and concatenates hidden representations obtained from aggregation function at each layer. These combined representations with an MLP layer are used for dimension reduction, yielding $\mathbf{h}_{modularity} = (\mathbf{h}_{G_1}, \ldots, \mathbf{h}_{G_T})$. We use 3 layers of MoE-GIN, and the embedding dimension in MoE-GIN is 128. For each graph $t$, the MoE-GIN is shared. Then, $\mathbf{h}_{modularity}$ is aggregated by the state experts to form $h_{state} \in \mathbb{R}^{K \times D}$. Each state feature with a shared MLP layer is used for final prediction and we average the results of each state as the final result. It is worth noting that the state experts are not exactly the same as MoEs.

## D  IMPLEMENTATION DETAILS

To obtain the results detailed in Section 4.2, we trained dFCExpert in an end-to-end supervised fashion using the following configuration:

- *Optimizer*: Adam
- *Learning rate*: A learning rate of $5e^{-4}$ was used to train the classification task, and a learning rate of $1e^{-3}$ was used to train the regression task.
- *Mini-batch size*: mini-batch size of 8 for the HCP dataset and mini-batch size of 16 for the ABCD dataset.
- *Epochs*: 50 epochs for training the HCP dataset, and 30 epochs for the ABCD dataset.

Table 3: Performance comparison with alternatives and baselines on HCP dataset.

| Method | HCP | | | | FC Type |
| | Sex | | Intelligence | | |
| | ACC(%) | AUC(%) | MSE($\downarrow$) | CORR($\uparrow$) | |
|---|---|---|---|---|---|
| BrainNetCNN Kawahara et al. (2017) | 84.56±1.87 | 91.97±1.33 | 1.130±0.039 | 0.200±0.071 | Static |
| BRAINNETTF Kan et al. (2022) | 85.42±2.36 | 92.51±2.01 | 1.015±0.052 | 0.205±0.074 | Static |
| NeuroGraph Said et al. (2023) | 84.38±2.60 | 92.03±2.39 | 1.321±0.043 | 0.198±0.065 | Static |
| STAGIN Kim et al. (2021) | 87.74±1.75 | 92.09±0.93 | 1.002±0.042 | 0.215±0.078 | Dynamic |
| MSGNN Wang et al. (2023) | 86.25±2.03 | 92.37±1.42 | 1.073±0.057 | 0.206±0.070 | Dynamic |
| NeuroGraph Said et al. (2023) | 84.66±3.30 | 93.14±2.23 | 1.017±0.032 | 0.207±0.069 | Dynamic |
| Baseline | 88.11±4.26 | 96.68±1.71 | 0.978±0.079 | 0.238±0.086 | Dynamic |
| Baseline + M | 89.61±4.22 | 96.91±1.68 | 0.946±0.041 | 0.272±0.059 | Dynamic |
| Baseline + S | 89.98±2.83 | 97.23±1.32 | 0.967±0.074 | 0.248±0.083 | Dynamic |
| dFCExpert | **91.03±1.71** | **97.40±1.30** | **0.932±0.036** | **0.296±0.049** | Dynamic |

Table 4: Performance comparison with alternatives and baselines on ABCD dataset

| Method | HCP | | | | FC Type |
| | Sex | | Intelligence | | |
| | ACC(%) | AUC(%) | MSE($\downarrow$) | CORR($\uparrow$) | |
|---|---|---|---|---|---|
| BrainNetCNN Kawahara et al. (2017) | 85.38±0.07 | 92.67±0.33 | 0.474±0.031 | 0.442±0.011 | Static |
| BRAINNETTF Kan et al. (2022) | 83.55±0.64 | 91.38±0.21 | 0.462±0.022 | 0.465±0.024 | Static |
| NeuroGraph Said et al. (2023) | 85.71±0.60 | 93.75±0.61 | 0.512±0.013 | 0.423±0.015 | Static |
| STAGIN Kim et al. (2021) | 86.93±0.45 | 91.57±0.57 | 0.452±0.012 | 0.477±0.008 | Dynamic |
| MSGNN Wang et al. (2023) | 87.13±0.63 | 92.67±0.42 | 0.453±0.024 | 0.483±0.017 | Dynamic |
| NeuroGraph Said et al. (2023) | 87.37±1.08 | 94.64±0.69 | 0.441±0.031 | 0.482±0.005 | Dynamic |
| Baseline | 87.25±0.70 | 94.86±0.70 | 0.442±0.021 | 0.484±0.013 | Dynamic |
| Baseline + M | 88.09±0.76 | 95.16±0.54 | 0.425±0.026 | 0.499±0.018 | Dynamic |
| Baseline + S | 88.56±1.25 | 95.81±0.53 | 0.419±0.014 | 0.496±0.013 | Dynamic |
| dFCExpert | **89.28±0.83** | **95.99±0.52** | **0.412±0.012** | **0.513±0.009** | Dynamic |

- *Number of experts*: $C = 7$ for HCP dataset and $C = 5$ for ABCD dataset
- *Number of states*: $K = 7$ for HCP dataset and $K = 5$ for ABCD dataset

Particularly, from our experiments and existing neuroscience studies, $C$ can be 5 or 7, and $K$ can be from 5 to 7. Any values for $C$ and $K$ in this range are reasonable and acceptable.

# E    PERFORMANCE COMPARISON WITH STANDARD DEVIATION

Here, we detail the standard deviation of the results posted in Table 1 (manuscript), which compares the performance of dFCExpert against other alternatives and baselines on both the classification and regression tasks. All of the experiments were performed using the 5 stratified data splits for each dataset, which was used across all models for a fair comparison. The above Table 3 and Table 4 show the performance posted in Table 1, and the following Table 5 show the performance posted in Table 2 along with the standard deviation among the 5 splits.

# F    MORE ABOUT MODULARITY EXPERTS

To further validate the effectiveness of the modularity experts, we perform more experiments, such as using MoE-GIN as the graph feature extractor in static FC methods, analyzing the scaling ratios $\alpha$ for the two auxiliary losses, analyzing the number of parameters, and also training MoE-GIN with labeled networks. If not specified, the experiments are performed on HCP dataset.

## F.1    MoE-GIN IN STATIC FC METHOD

We implement our MoE-GIN scheme for static FC analysis, similar to Neurograph Said et al. (2023) that implements a GNN architecture to incorporate residual connections and concatenates hidden representations obtained from message passing at each layer. Finally, these combined representations with an MLP layer are used for predictions. As demonstrated by the results summarized in Table 6 , characterizing brain modularity scheme in static FC analysis can also improve the brain graph representations and achieve better performance, especially on the HCP dataset.

Table 5: Performance of the number of experts used in the modularity experts.

| #Experts | HCP | | | | ABCD | | | |
|---|---|---|---|---|---|---|---|---|
| | Sex | | Intelligence | | Sex | | Cognition | |
| | ACC(%) | AUC(%) | MSE(↓) | CORR(↑) | ACC(%) | AUC(%) | MSE(↓) | CORR(↑) |
| 1 (baseline) | 88.11±4.25 | 96.68±1.71 | 0.978±0.079 | 0.238±0.086 | 87.25±0.70 | 94.86±0.70 | 0.442±0.021 | 0.484±0.013 |
| 3 | 88.68±2.57 | 96.46±1.58 | 0.961±0.067 | 0.266±0.061 | 87.57±0.94 | 95.16±0.70 | 0.435±0.019 | 0.481±0.005 |
| 5 | 89.25±3.46 | 96.41±1.27 | 0.953±0.054 | **0.279±0.042** | **88.09±0.76** | **95.16±0.54** | **0.425±0.026** | **0.499±0.018** |
| 7 | **89.61±4.22** | **96.91±1.68** | **0.946±0.041** | 0.272±0.059 | 87.80±1.06 | 94.85±0.65 | 0.429±0.023 | 0.481±0.016 |
| 9 | 88.73±3.48 | 95.26±2.10 | 0.963±0.036 | 0.224±0.051 | 87.65±0.87 | 94.98±0.69 | **0.425±0.020** | 0.479±0.010 |
| 17 | 87.83±3.83 | 95.87±1.47 | 0.976±0.032 | 0.209±0.069 | 87.19±1.30 | 94.84±0.53 | 0.431±0.024 | 0.477±0.020 |

Table 6: Performance of MoE-GIN used in **static-FC** method.

| Method | GIN Type | HCP | | | | ABCD | | | |
|---|---|---|---|---|---|---|---|---|---|
| | | Sex | | Intelligence | | Sex | | Cognition | |
| | | ACC(%) | AUC(%) | MSE(↓) | CORR(↑) | ACC(%) | AUC(%) | MSE(↓) | CORR(↑) |
| NeuroGraph (Said et al., 2023) | GIN | 84.38±2.60 | 92.03±2.39 | 1.321±0.043 | 0.198±0.065 | 85.71±0.60 | 93.75±0.61 | 0.512±0.013 | 0.423±0.015 |
| NeuroGraph (Said et al., 2023) | MoE-GIN | 87.30±1.33 | 95.56±1.68 | 0.989±0.035 | 0.243±0.052 | 87.08±0.74 | 93.12±0.52 | 0.486±0.021 | 0.477±0.017 |

## F.2 Scaling Ratio $\alpha$ Analysis for Auxiliary Losses

Then, we analyze different values of the scaling ratio $\alpha$ for the two auxiliary losses in the modularity experts, and we experiment with $\alpha \in \{0.1, 1\}$ on the HCP dataset. The results are summarized in Table 7. We can see that, using $\alpha = 0.1$ resulted in improved performance on both tasks, indicating the importance of two auxiliary losses for balance loading and sparse gating. When $\alpha = 1$, better classification performance is achieved but the improvement on the regression task is not significant. We set $\alpha = 1$ in all our experiments. Further, to show the effectiveness of the proposed sparse gating loss $L_{sparse}$, we also ran experiments of setting the scaling ratio of $L_{sparse}$ to 0 (while the ratio of $L_{balance}$ is 1). We find that without sparse gating to enable the specialization of experts, the performance is decreased.

Table 7: Results for modularity experts with different scaling ratios $\alpha$ ($\alpha_b$ and $\alpha_s$ are the scaling ratios for $L_{balance}$ and $L_{sparse}$, respectively).

| $\alpha$ | $\alpha_b, \alpha_s$ | Sex | | Intelligence | |
|---|---|---|---|---|---|
| | | ACC(%) | AUC(%) | MSE(↓) | CORR(↑) |
| 0 | $\alpha_b = 0, \alpha_s = 0$ | 88.56±3.45 | 94.67±1.68 | 0.962±0.022 | 0.256±0.048 |
| 0.1 | $\alpha_b = 0.1, \alpha_s = 0.1$ | 89.16±4.12 | 95.52±2.01 | 0.948±0.032 | 0.270±0.036 |
| 1 | $\alpha_b = 1, \alpha_s = 1$ | 89.61±4.22 | 96.91±1.68 | 0.946±0.041 | 0.272±0.059 |
| - | $\alpha_b = 1, \alpha_s = 0$ | 87.83±4.05 | 94.84±2.19 | 0.965±0.028 | 0.252±0.050 |

## F.3 Analysis for Number of Parameters

One may argue that the performance gain of our modularity expert is due to the increasing model parameters, since each expert has with its own trainable parameters. This could be one of the reasons, but the MoE concept originally is designed to improve the model capacity, and it is also a good way to learn representations with diverse graph structures, such as the fMRI data. Further, in Table 2, we also observe that increasing the number of experts (i.e., model parameters) does not necessarily lead to performance improvements, especially on the ABCD dataset. When $C = 3$, the performance is even lower than the baseline method on the regression task.

Additionally, we perform more experiments to validate the effectiveness of our method. Particularly, we increase the number of parameters of baseline method to reach the same level as our modularity experts. The difference in the number of parameters between the baseline and our modularity experts lies in the number of experts used in the graph layer. For example, if the number of experts is $C = 7$, the number of parameters in MoE-GIN of our method can be 7 times of those in GIN of baseline method. To reach similar level for the number of parameters, we set the feature embedding dimension of the GIN layers in baseline method to 384. The results are illustrated in Table 8. We find that increasing the number of parameters can improve the performance to some extent ("Baseline(384)"), but our modularity scheme still performs better, especially on the regression task. This shows that the performance gain of our method is not entirely from the increased parameters, but also from the modularity scheme to characterize the brain modularity in the graph learning process, thus effectively learning graph representations.

Table 8: Analysis for the number of parameters of modularity experts. "Baseline(128)" is the baseline method that the feature embedding dimension of the GIN layers is 128, similarly for "Baseline(128) + M" and "Baseline(384)", where "M" indicates our modularity scheme.

| Method | #Param. | Sex | | Intelligence | |
|---|---|---|---|---|---|
| | | ACC(%) | AUC(%) | MSE($\downarrow$) | CORR($\uparrow$) |
| Baseline(128) | 20.87M | 88.11±4.26 | 96.68±1.71 | 0.978±0.079 | 0.238±0.086 |
| Baseline(128) + M | 21.69M | 89.61±4.22 | 96.91±1.68 | 0.946±0.041 | 0.272±0.059 |
| Baseline(384) | 21.83M | 87.66±3.89 | 95.72±1.54 | 0.963 ±0.032 | 0.258±0.042 |

### F.4 USING LABELS WHEN TRAINING MoE-GIN

According to the Schaefer atlas, we can use network labels to assign the functional brain network nodes to different brain networks (modules). To verify whether our method can allow each expert to specialize in specific brain modules, we compare the performance of our modularity method trained with the two auxiliary losses (unsupervised) or cross-entropy (CE) loss with labels (supervised). As summarized in Table 9, the results clearly indicate that the proposed auxiliary losses and the labeled CE loss achieve similar performance on the sex classification task, and on the regression tasks our proposed auxiliary losses perform better. These results confirm the effectiveness of the two auxiliary losses for unsupervised clustering, and also indicates that the modularity experts can learn the functional modules beyond the predefined atlas, having great potential to capture the personalized networks.

Table 9: Performance comparison of training MoE-GIN using network labels or our proposed loading balance and sparse gating loss.

| Method | Loss | Sex | | Intelligence | |
|---|---|---|---|---|---|
| | | ACC(%) | AUC(%) | MSE($\downarrow$) | CORR($\uparrow$) |
| Baseline + M | labeled CE loss | 89.55±4.38 | 96.98±1.23 | 0.968±0.033 | 0.258±0.027 |
| Baseline + M | $L_{balance}$ and $L_{sparse}$ | 89.61±4.22 | 96.91±1.68 | 0.946±0.041 | 0.272±0.059 |

## G MORE ABOUT STATE EXPERTS

In this section, we present more experimental results about the state experts, including different prototype center initialization methods, the loss trade-off weight $\beta$ for $L_{state}$, and visualization of FC differences between males and females. All the experiments are performed on HCP dataset.

### G.1 SCALING FACTOR $\beta$ ANALYSIS FOR $L_{state}$

Further, we conduct an investigation into the scaling factor for the loss $L_{state}$. As shown in Table 10, $\beta = 1$ yields improved performance compared to $\beta = 0$. This indicates the crucial role of implementing loss $L_{state}$ for state patterns learning. Since the loss $L_{state}$ is relatively small, to better exert it's role we also test the performance when $\beta = 10$. The resulting performance is further improved, showing better ability for enhancing the state assignment.

Table 10: Results for state experts with different scaling ratios $\beta$.

| $\beta$ | Sex | | Intelligence | |
|---|---|---|---|---|
| | ACC(%) | AUC(%) | MSE($\downarrow$) | CORR($\uparrow$) |
| 0 | 86.75±2.43 | 93.52±1.12 | 1.023±0.065 | 0.210±0.072 |
| 1 | 88.23±2.12 | 96.28±1.85 | 0.974±0.044 | 0.268±0.063 |
| 10 | 89.98±2.83 | 97.23±1.32 | 0.967±0.074 | 0.248±0.083 |

### G.2 MORE VISUALIZATION RESULTS

As a supplement to the results of main paper in Section 4.4, we further visualize the averaged FCs of temporal segments in males or females assigned to each state. As the results shown in Figure G.2, the state experts indeed can capture different dynamic patterns of dFC measures. The sex difference of these averaged state FC measures also highlights the interpretability of state experts. For example,

dFC graphs (networks) of females may have strong activities at state 2, and also states of 1, 4, 7, while males are more active at states of 3, 5, and 6. Additionally, we can also observe different FC strengths between males and females.

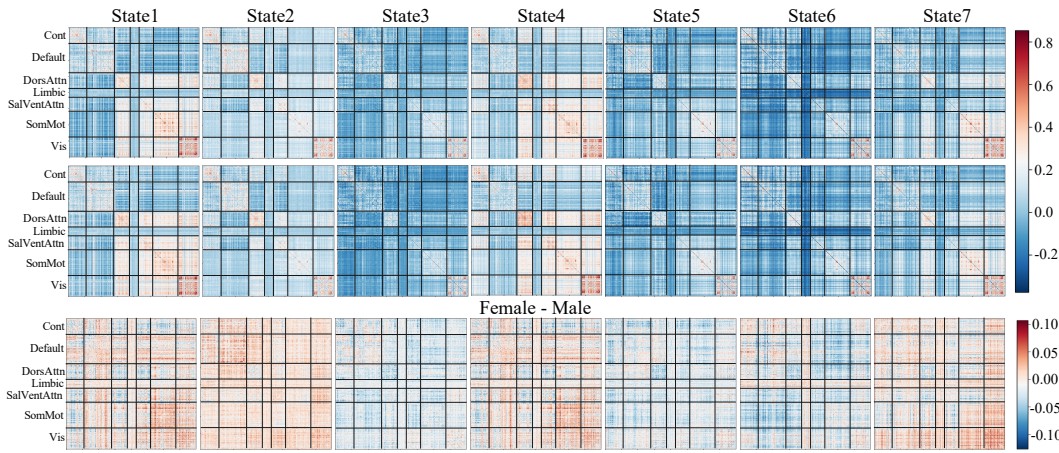

Figure 5: Visualization of averaged FCs of temporal segments of males and females allocated to a specific state. The first row is the female group, second row is the male group, and the third row shows FC differences (female - male) of female and male groups in each state. The y-axis indicates the 7 networks: frontoparietal, default mode, dorsal attention, limbic, ventral attention, somatomotor and visual networks.

