# OpenReview forum: "dFCExpert: Learning Dynamic Functional Connectivity Patterns with Modularity and State Experts"
_ICLR.cc/2025/Conference — ICLR 2025 Conference Withdrawn Submission_

### Official Review · Reviewer_GWjL · 2024-11-02

**Soundness:** 2
**Presentation:** 3
**Contribution:** 2
**Rating:** 3
**Confidence:** 4

**Summary:**

This paper introduces dFCExpert, a method aimed at learning effective representations of dynamic functional connectivity patterns from functional magnetic resonance imaging (fMRI) data. The approach combines Graph Neural Networks (GNNs) with Mixture of Experts (MoE) models, featuring two essential components: modularity experts and state experts. Experimental results show that dFCExpert significantly surpasses existing methods in both gender classification and cognitive intelligence prediction tasks, while also improving the model's interpretability.

**Strengths:**

1. The dFCExpert method integrates modular and state experts, representing the first incorporation of the brain's modular properties and dynamic state mechanisms into fMRI data analysis, offering a new perspective for future research.
2. dFCExpert outperformed existing methods in both sex classification and cognitive intelligence regression tasks, showcasing its effectiveness.

**Weaknesses:**

1. While dFCExpert combines GNN, MoE, and soft clustering, these techniques are well-established in prior research. The paper's contribution is in the integration of these methods, rather than proposing a fundamentally new model architecture, which limits the overall innovation. If there are any novel modifications or adaptations made to the GNN, MoE or clustering components to tailor them to brain connectivity data.

2. While combining different expert modules enhances the model's performance, this concatenation strategy may increase the complexity of hyperparameter tuning and pose a risk of optimization instability. Additionally, this approach may yield inconsistent results across different datasets or tasks, potentially affecting the method's generalizability. So the authors may need to conduct evaluations of the hyperparameter sensitivity, optimization stability, and cross-dataset/task generalizability of their approach.

3. While the authors describe the method clearly, there is a lack of explanation for certain key decisions, such as the rationale for selecting GIN from multiple GNN variants and the choice of the soft prototype clustering method. Providing a justification for these selections or presenting comparative experiments would strengthen the paper.

4. Although two authoritative large-scale datasets were used, the scope of the experimental validation remains limited. The study lacks extensive experiments across different brain disorder datasets or diverse neuroimaging datasets, making it difficult to comprehensively assess the method's generalization capability. It remains unclear whether the model can perform well across a variety of brain states. The authors could evaluate their method on additional datasets to demonstrate broader generalizability. For example, commonly used datasets such as ABIDE I, ABIDE II, ADNI, and ADHD, etc. The portion of HCP data used by the authors is only a small part, and the ABCD data is also insufficient to support their research.

5. The authors state that the criterion for selecting downstream tasks is to 'explore fundamental brain-biology and brain-cognition associations.' However, the choice of the sex classification task does not seem to align well with this standard, making the task selection appear somewhat arbitrary. The authors should consider alternative downstream tasks that may better align with exploring brain-biology and brain-cognition associations.

6. In the MODULARITY EXPERTS ANALYSIS section, the authors state, 'From our results, C can be 5 or 7, which are both reasonable and also consistent with the typical number of functional modules in neuroscience.' However, this claim lacks support from neuroscience literature. It is recommended to provide relevant citations to substantiate this statement.

7. In the STATE EXPERTS ANALYSIS section, the authors compare the performance differences with varying numbers of state experts but do not provide a thorough and well-reasoned analysis of the experimental results. It is recommended to include a more in-depth interpretation to better understand the impact of the number of state experts on the model's performance.

8. The experiments in the paper should also consider imaging from different brain region templates, such as AAL 90/116, CC200, and BNA246.

**Questions:**

1. While dFCExpert combines GNN, MoE, and soft clustering techniques that are well-established in prior research, could the authors clarify if there are any novel modifications or adaptations made to the GNN, MoE, or clustering components specifically tailored to brain connectivity data? How do these adaptations contribute to the overall innovation of the model?

2. The concatenation strategy used to combine different expert modules may increase the complexity of hyperparameter tuning and pose a risk of optimization instability. How do the authors plan to evaluate the hyperparameter sensitivity, optimization stability, and cross-dataset/task generalizability of their approach to address these concerns?

3. Although the authors describe the method clearly, could they provide a justification for certain key decisions, such as the rationale for selecting GIN from multiple GNN variants and the choice of the soft prototype clustering method? Would presenting comparative experiments strengthen the paper's claims?

4. The experimental validation seems limited despite using two authoritative large-scale datasets. Could the authors elaborate on their plans to conduct extensive experiments across different brain disorder datasets or diverse neuroimaging datasets to comprehensively assess the method's generalization capability? How do they intend to demonstrate that the model can perform well across a variety of brain states, possibly by evaluating on commonly used datasets such as ABIDE I, ABIDE II, ADNI, and ADHD? Given that the portion of HCP data used is only a small part, do they recognize the insufficiency of ABCD data to support their research?

5. The authors state that the criterion for selecting downstream tasks is to "explore fundamental brain-biology and brain-cognition associations." However, does the choice of the sex classification task truly align with this standard, or does it appear somewhat arbitrary? Could the authors consider alternative downstream tasks that may better align with exploring brain-biology and brain-cognition associations?

6. In the MODULARITY EXPERTS ANALYSIS section, the authors claim that "C can be 5 or 7, which are both reasonable and consistent with the typical number of functional modules in neuroscience." Could the authors provide relevant citations from neuroscience literature to substantiate this statement?

7. In the STATE EXPERTS ANALYSIS section, while the authors compare performance differences with varying numbers of state experts, could they include a more thorough and well-reasoned analysis of the experimental results? How does the number of state experts impact the model's performance?

8. Should the experiments in the paper also consider imaging from different brain region templates, such as AAL 90/116, CC200, and BNA246? How might incorporating these templates enhance the study's findings?

---

### Official Review · Reviewer_npnm · 2024-11-03

**Soundness:** 3
**Presentation:** 4
**Contribution:** 3
**Rating:** 6
**Confidence:** 4

**Summary:**

This paper proposes dFCExpert, an MoE-based learning framework for dynamic functional connectivity in terms of modularity (space) and state (time). The proposed methodology is acceptable and neuroscientifically plausible, showing strong experimental results on two large-scale datasets. The paper is well-written and the demonstration is easy to follow. Some concerns and suggestions are outlined in the `Questions` section.

**Strengths:**

- Acceptable methodology with a neuroscientifically plausible approach
- Strong experimental results on benchmark performance and analyses
- Well-written demonstration of the work which is easy to follow

**Weaknesses:**

- Interpretation of the two expert outputs can be further elaborated

**Questions:**

### Major concerns
- Please consider adding an interpretation of each state and what can be learned from the difference between the female and male state assignment pattern in Section 4.4 (Figure 4). Appendix G.2 visualizes the functional connectivity of each state and the difference between the two sexes, but it is still difficult to interpret the neuroscientific implication because each matrix is small and the colorbar is not centered to zero.
- Please consider adding a subfigure that color-codes the modularity-matched ROIs versus unmatched ROIs with the Schaefer 7 networks. It is not straightforward to see which areas of the brain show alignment with the Schaefer 7 networks in its current form.
- Please clarify which atlas was used for the experiments along with the number of ROIs.

### Minor concerns
- Extending the experiments to a clinical fMRI dataset (e.g., ABIDE) would be of interest to the readers in the field.
- The typefaces that denote functions, vectors, and matrices are often inconsistent, which lowers readability. Please consider revising the typefaces of notations to be consistent throughout the paper. For example, $\mathbf{P}$ is a matrix, while $\mathbf{G}$ is a set. $T$ is a scalar, $X_{t}$ is a matrix, and $F_{a}$ is a function, and so on.
- Typo: The S"c"haefer atlas is often referred to as the Shaefer atlas.


### Recommendation
I find the strengths of this work outweigh the above concerns. Thus, I recommend accept of the paper.

---

### Official Review · Reviewer_e9U6 · 2024-11-03

**Soundness:** 3
**Presentation:** 2
**Contribution:** 1
**Rating:** 3
**Confidence:** 4

**Summary:**

In this work, the authors have proposed a novel analytical tool for deriving modularized dynamical functional connectivity patterns, via modularity and state experts. In the neuroscience field, quasi-periodic property of brain dynamics has been of great interest; thus, have been studied using different modalities including functional MRI. Here, the authors suggested two tacks of model; the modularity expert, implemented using graph neural network (GNN) and mixture of experts, learns graph-like network patterns whereas the state expert reveals brain states given graph-based features. The major scientific novelty of this work comes from its well-designed analytical schematic for dynamic functional connectivity.

**Strengths:**

Based on previous studies, here the authors have proposed a novel analytical tool that can be interesting to neuroscientists. Especially, the conceptual advance proposed in this work clearly shows the originality of this work. The validity of proposed method has been extensively demonstrated throughout the work, showing its high quality. Plus, the details of the proposed model are well described; guaranteeing the clarity of this work.

**Weaknesses:**

While there are several strengths in this work, there are some weaknesses worth mentioning. First, although the scientific novelty of the proposed model is clear, it is very unclear whether this model can be practically more useful than other previously proposed models. Given high standard deviation, it seems like the performance achieved by the proposed method is marginally better than other models. Throughout the manuscript, the authors did not present any results demonstrating unique aspects or information of the proposed model. This weakness greatly undermines the scientific significance of this work.

**Questions:**

1.	To my understanding, C and K, # of modularity experts and state experts, are critical hyperparameters of the proposed model. I assume those two parameters were somehow optimized. In Table 1, what were C and F of dFCExpert? Given results in Table 2 and Figure 4, I feel like the performance of dFCExpert is comparable to other models. Does it suggest that the superior performance of dFCExpert largely comes from its optimized hyperparameter rather than conceptual advances?

Below are specific questions related to it:

- Please clarify C and K values used for dFCExpert in Table 1
- Please describe hyperparameter optimization process in Table 1 and other results. In other words, how do we know this optimized parameter set not overfitted to the specific datasets?
- Please provide results comparing dFCExpert to other models when using similar hyperparameter settings. This would help distinguish between gains from conceptual advances versus hyperparameter tuning.

2.	As stated in weakness, are there any unique aspects that can be achieved by the proposed model? Observations in Figure 4 are already reported with conventional methods (dFC + K-mean clustering). In addition to increment in accuracy, I suggest the authors perform additional analysis to support the unique strength of the proposed model.

- Please compare the interpretability of learned states between dFCExpert and conventional methods.
- Demonstrate the underlying mechanism how dFCExpert captures individual differences in brain modularity.
- Analyze the stability of identified states across different runs or subsets of data.

3.	The authors reported different C/K between HCP and ABCD datasets. I suspect this inconsistency simply suggest that the proposed method is unstable. If the authors think this discrepancy in C and K reflects important neurophysiological difference between two datasets, the authors should further support such results.

- Explain the author's rationale why using different C/K values for each dataset.
- Provide results using consistent C/K values across both datasets to demonstrate stability
- If the authors believe the different values reflect neurophysiological differences in datasets, please consider to provide additional results to support this claim.

---

### Official Review · Reviewer_Xuyn · 2024-11-06

**Soundness:** 1
**Presentation:** 3
**Contribution:** 1
**Rating:** 1
**Confidence:** 5

**Summary:**

The paper presents a MoE scheme for GNN, aiming at improving the dynamic functional connectivity analysis for fMRI data. The major technical component of this work is implemented through 1) Modularity Expert, which essentially learns an MoE at each graph layer, capturing nodes with similar functions (e.g., neighboring). The modularity experts use a gating function to determine which experts to use for a given node, regularized by a sparsity loss. The function is implemented so that each node will only route to a single expert. 2) State Expert, which performs soft prototype clustering to aggregate temporal graph features into connectivity states. The model was tested on HCP and ABCD datasets and evaluated by gender classification and cognitive intelligence regression tasks. Experiment results show improved performance compared to other graph-based methods for fMRI analysis.

**Strengths:**

Allowing sparsity-regularized modeling through MoE for graph-based analysis is a useful feature, especially considering the growing utilization of MoE in the field.

**Weaknesses:**

First and foremost, it is unclear how/why the modularity expert is effective in the proposed model. Specifically, there is no theoretical analysis in 3.2.2 on why introducing the MoE-GIN layer will help, nor why the current gating function is selected for this task. In addition, Table 1 shows only marginal improvement for incorporating the modularity expert over the baseline (3-layer GIN + MLP) method. Fig. 3 is not helping either, and the author’s statement “The visualization clearly demonstrated that the Schaefer atlas networks were highly overlapped with the modules identified by the modularity experts, indicating that the modularity experts could assign tightly connected nodes to the same experts (module) to allow each expert to capture nodes of one brain functional module” is not convincing as the 7-network atlas is way too simple to observe any meaningful brain segregation, especially considering the fact that the MoE is designed to characterize neighboring nodes.

Secondly, the prototype clustering method used by the state experts appears to be an ad-hoc addition without justification. The authors don't explain why this particular clustering approach was chosen over other alternatives, nor do they provide evidence that the learned states correspond to meaningful brain states. Similar to the issue above, the improvement from adding state experts (Baseline + S in Table 1) is minimal. The result doesn't justify the additional model complexity and computational overhead, which could be the real reason for the improvement in performance.

**Questions:**

Will the MoE scheme work for other GNN architectures as well?

**Details Of Ethics Concerns:**

No ethics concerns. Only public data were used in this work and the functionality of the proposed model is not potentially harmful.

---

### Note · Authors · 2024-11-23

I have read and agree with the venue's withdrawal policy on behalf of myself and my co-authors.